# Discrepancy in Sterol Usage between Two Polyphagous Caterpillars, *Mythimna separata* and *Spodoptera frugiperda*

**DOI:** 10.3390/insects13100876

**Published:** 2022-09-27

**Authors:** Rui Tang, Junhao Liang, Xiangfeng Jing, Tongxian Liu

**Affiliations:** Key Laboratory of Northwest Loess Plateau Crop Pest Management of Ministry of Agriculture and Rural Affairs, Northwest A&F University, Yangling, Xianyang 712100, China

**Keywords:** *Mythimna separata*, *Spodoptera frugiperda*, cholestanone, cholesterol, metabolism, nutrition

## Abstract

**Simple Summary:**

*Mythimna separata* and *Spodoptera frugiperda* are two destructive pests worldwide. In this study, for the first time, we evaluated their sterol metabolic capacity. The results showed that *Spodoptera frugiperda* required less sterol for normal growth, which helped them survive better when cholesterol was unavailable. Both insects consistently showed high fitness when they fed on cholestanol. Cholestanone enabled most individuals of *S*. *frugiperda* to pupate but caused remarkable lethality to *M*. *separata* at their early developmental stage. Comparative studies indicated that *S*. *frugiperda* was more efficient in converting ketone into available stanol than *M*. *separata*. Therefore, they perform differently in terms of their sterol demand and metabolism although these two species are closely related. The divergences in sterol nutritional biology between the two closely related insect species reflect adaptive and evolutive changes in sterol metabolism, which may help us to better understand the potential of using phytosterol-manipulated plants to control pests.

**Abstract:**

Insects are sterol auxotrophs and typically obtain sterols from food. However, the sterol demand and metabolic capacity vary greatly among species, even for closely related species. The low survival of many insects on atypical sterols, such as cholestanol and cholestanone, raises the possibility of using sterol-modified plants to control insect herbivore pests. In this study, we evaluated two devastating migratory crop pests, *Mythimna separata* and *Spodoptera frugiperda*, in response to atypical sterols and explored the reasons that caused the divergences in sterol nutritional biology between them. Contrary to *M*. *separata*, *S*. *frugiperda* had unexpectedly high survival on cholestanone, and nearly 80% of the individuals pupated. Comparative studies, including insect response to multiple diets and larval body sterol/steroids analysis, were performed to explain their differences in cholestanone usage. Our results showed that, in comparison to *M*. *separata*, the superiority of *S*. *frugiperda* on cholestanone can be attributed to its higher efficiency of converting ketone into available stanol and its lower demand for sterols, which resulted in a better survival when cholesterol was unavailable. This research will help us to better understand insect sterol nutritional biology and the potential of using atypical sterols to control herbivorous insect pests.

## 1. Introduction

Lepidoptera comprises the second most diverse insect order, and many species are notorious pests that can cause gigantic yearly economic losses [1,2,3,4]. The oriental armyworm, *Mythimna separata* (Walker), and the fall armyworm, *Spodoptera frugiperda* (J.E. Smith), are two noctuid pests that are found worldwide. Both pests are well-known long-distance migrators with high reproductive efficiency [5,6]. Practically, effective management of these pests heavily relies upon chemical insecticides, and high resistance has been reported [7,8,9,10]. Therefore, environmentally friendly strategies are urgent for controlling these pests.

Synthesizing cholesterol de novo is metabolically expensive, and insects gradually discard the genes involved in cholesterol synthesis [11,12,13]. However, cholesterol, as the dominant sterol in most insect species, participates in several key physiological processes, including, but not limited to, (1) cell-to-cell recognition, adhesion, and communication as a membrane structural component [14,15]; (2) metamorphosis as the precursor of steroid hormones [16,17]; (3) protection against pathogenic agents and parasitoids [18,19]; and (4) organismal growth and patterning as the signal molecules via the Hedgehog signaling pathway [20]. As a result, insects do require an exogenous cholesterol source for normal growth and development [13]. Cholesterol is rarely found in plants above trace levels, so insect herbivores have to convert phytosterols into cholesterol [21]. Lepidopterans seem to possess a robust capacity for metabolizing a variety of typical phytosterols, such as sitosterol and stigmasterol, to cholesterol by dealkylation [21,22]. More than 200 different types of phytosterols have been discovered in plants, and there are some atypical sterols at varied levels in plants, such as ketone and stanol [23,24]. These atypical sterols/steroids are not readily metabolized to cholesterol by insects and even interfere with the normal function of cholesterol to the point that they can be deleterious. For example, all the *Helicoverpa zea* larvae and more than 60% of *Heliothis virescens* larvae died when they were reared on cholestanone [25]. Similar to vertebrates, dietary sterol uptake in insects seems to be a non-selective process, so insects can be significantly affected by dietary sterol composition [21,26,27]. These sterol–metabolic limitations can be exploited to develop non-insecticidal strategies. For example, increasing the proportion of unsuitable sterols in plants or inhibiting metabolic targets were proved to be promising for controlling phytophagous pests [28,29,30].

Insects are discrepant in dietary sterol/steroid usage and demand, even between closely related species [31,32]. Both *S. frugiperda* and *M*. *separata* are migratory noctuid insects with similar life histories [1,2]. *Mythimna separata* larvae mostly feed on grain crops. In contrast, in addition to grain crops, *S*. *frugiperda* larvae can feed on many other economically important plants, e.g., beet, tomato, potato, and cotton [33,34]. Therefore, *S*. *frugiperda* seems to have stronger adaptabilities and likely encounters more types of sterols than *M*. *separata*. Here, we propose that *S. frugiperda* has stronger capabilities in the use of sterols than *M. separata*. In this study, we focus on the metabolic discrepancies of atypical sterols—which were used to increase plant resistance to lepidopteran pests—between two gluttonous pests, *M*. *separata* and *S*. *frugiperda*, and we also examine the difference in sterol demand between these two species.

## 2. Materials and Methods

### 2.1. Insect Culture and Artificial Diet

We purchased *M. separata* and *S*. *frugiperda* pupae from Keyun Industry (Jiyuan, Henan, China). Newly emerged adults were reared on 10% honey solution (vol/vol). Eggs produced by these adults were allowed to hatch. All larvae were individually reared on an artificial diet in 24-well cell culture plates and kept in an incubator at 28 ± 1 °C, 75 ± 5% RH, with an L16:D8 photoperiod. Semi-synthetic diets were prepared according to the procedure described previously [35]. Food that slightly exceeded daily consumption was provided and refreshed daily. Cholesterol (C, CAS: 57-88-5, >99%) and cholestanol (A, CAS: 80-97-7, >97%) were purchased from Sigma Chemical (St. Louis, MO, USA), and cholestanone (K, CAS: 566-88-1, >98%) was purchased from Steraloids Inc. (Newport, RI, USA). The chemical structures of the sterols/steroids involved in this study are shown in Figure 1.

### 2.2. Growth Requirement of Cholesterol

Two separate experiments were performed. In one experiment, five artificial diets that only differed in their cholesterol concentrations (dry mass) were prepared: NC (0 mg/g), 0.25C (0.25 mg/g), 0.5C (0.5 mg/g), 0.75C (0.75 mg/g), and 1C (1 mg/g). Twenty-four newly hatched larvae of *M*. *separata* and *S*. *frugiperda* were randomly assigned to these five diets. In the other experiment, newly hatched larvae from both species were reared on a cholesterol diet (1 mg/g) after they hatched. Insects were checked daily. Food was removed when larvae were about to molt to the 4th instar, which was determined by head capsule slippage (Appendix A). Then, a new cholesterol diet that slightly excessed daily consumption was applied to each insect daily, and the quality was recorded. The remaining food was collected daily and dehydrated in a baking oven before weighing. A regression line for each diet was created to calculate the initial dry mass of the food given to the insects (Appendix A). Food consumption by each insect was calculated as the difference between the initial dry mass and the dry mass of the remaining food. Once they molted to the 6th instar, the larvae were freeze-dried and weighed for sterol/steroid analysis by GC/MS technology [36]. Sixteen individuals were used to calculate the food consumption for each treatment. Five biological samples were prepared for sterol analysis, and each sample consisted of three individuals.

### 2.3. Larval Response to Atypical Sterol Diets

Upon hatching, the neonates of *M*. *separata* and *S*. *frugiperda* were offered a cholestanol or cholestanone diet at a concentration of 1 mg/g (dry mass). Cholesterol at the same concentration was used as the control. Insects were checked daily for death and pupation, and the pupal weight was measured on the second day after pupation. Twenty-four larvae were used for each treatment.

### 2.4. Larval Performance on Cholestanone

Due to the high mortality of *M*. *separata* larvae on the cholestanone diet, another experiment was performed. Neonates of both species were reared on a cholesterol diet (1 mg/g), and the insects were checked daily. Upon molting to the 4th instar, larvae were weighed and transferred onto a diet containing 1 mg/g cholestanone or cholesterol. Food consumption by each insect during the 4th and 5th instars was calculated using the method mentioned above. Weight gain was calculated as the difference between the initial weight at the 4th instar and the initial weight at the 6th instar. Relative food intake was calculated as the ratio of the food intake of each insect on cholestanone to the mean of the food intake of the insects on cholesterol. Relative weight gain was calculated by the ratio of the weight gain of each insect on cholestanone to the mean of the weight gain of the insects on cholesterol. Once they molted to the 6th instar, larvae on cholestanone were weighed and freeze-dried for sterol/steroid analysis. Sixteen larvae from each species for each diet were used to calculate relative food intake and relative weight gain. Eight biological samples were prepared for the sterol/steroid analysis for both species, and each sample consisted of three individuals.

### 2.5. Statistical Analyses

Statistical analyses were performed using GraphPad Prism 8 software (GraphPad Software, San Diego, CA, USA). The survival of *M. separata* and *S*. *frugiperda* on cholesterol diets were analyzed by Kaplan–Meier procedure and log-rank tests, respectively. The differences in the effects of different sterols on the two insects were analyzed by one-way ANOVA with Tukey’s test. The significant differences between the two samples were analyzed using Student’s *t*-test. Differences were considered significant at *p* < 0.05. Values were presented as means ± standard errors.

## 3. Results

### 3.1. Growth Requirement of Cholesterol

To determine the dietary cholesterol concentration suitable for the survival and development of both *S*. *frugiperda* and *M*. *separata*, we evaluated the larval response to a series of diets that contained different cholesterol concentrations. No insect died on the diet that contained 1 mg/g cholesterol (1C diet), and the performance significantly decreased when the concentration of dietary cholesterol was reduced (*p* < 0.0001; Figure 2). We terminated the experiment on the 12th day when all 0.25C-fed *M*. *separata* larvae died. Compared with the insects on the 1C diet, the survival rate of *M*. *separata* larvae on the 0.75C diet was significantly lower on the 12th day, while it was not significantly different for the larvae of *S*. *frugiperda* (Figure 2). Only 25% of *M*. *separata* larvae survived on the 0.5C diet, and none survived on the 0.25C diet. In contrast, the 0.5C and 0.25C diets allowed 75% and 8.3% of *S*. *frugiperda* larvae to reach the 12th day, respectively (Figure 2). Furthermore, a large number of *M*. *separata* larvae (87%) on the 0.75C diet and all larvae on the 0.5C diet stagnated at the 1st or 2nd instar, but 91% (on the 0.75C diet) and 28% (on the 0.5C diet) of *S*. *frugiperda* larvae developed into the 3rd instar, respectively (Figure 3).

To quantify the sterol demand of *M*. *separata* and *S*. *frugiperda*, we measured the feeding amount and body sterol profiles of the larvae that fed on cholesterol (1 mg/g) during the 4th and 5th instars. We found that *M*. *separata* larvae consumed much more food than *S*. *frugiperda* (*t*_30_ = 6.28, *p* < 0.0001; Figure 4a), and the concentration of cholesterol was significantly higher in the bodies of *M*. *separata* than in *S. frugiperda* (*t*_8_ = 4.00, *p* = 0.004; Figure 4b).

### 3.2. Effects of Atypical Steroids on Larval Performance

Compared with cholesterol, the larval performance of both *M*. *separata* and *S*. *frugiperda* was significantly decreased when the insects fed on cholestanol or cholestanone (Table 1). All larvae of the two insects on cholestanol pupated. However, the developmental time was delayed, and the pupa weight was lighter than those on cholesterol. The larval performance of *M*. *separata* on cholestanone was significantly poorer than those fed on cholestanol. No larvae that fed on cholestanone pupated, and all larvae died at their early developmental stage, i.e., at the 1st or 2nd instar. By contrast, nearly 80% of cholestanone-fed *S*. *frugiperda* larvae pupated, and the larval developmental time and pupal mass were comparable to those on cholestanol (developmental time: *t*_46_ = 1.76, *p* = 0.09; pupal mass: *t*_46_ = 1.10, *p* = 0.28; Table 1).

### 3.3. Larval Response to Cholestanone

As a remarkably different performance was observed between *S*. *frugiperda* and *M*. *separata* on cholestanone, we further measured the differences in metabolizing cholestanone between *S*. *frugiperda* and *M*. *separata*. We fed both species a cholesterol diet (1 mg/g) from the 1st instar. One half of the newly molted 4th instar larvae were then transferred onto the cholestanone diet, and the other half were transferred onto the cholesterol diet. At the end of the experiment, we found that *S*. *frugiperda* larvae had a significantly higher relative food consumption (*t*_30_ = 8.29, *p* < 0.0001; Figure 5a) and weight gain (*t*_30_ = 4.62, *p* < 0.0001; Figure 5b) than *M*. *separata* during the cholestanone-feeding period. GC/MS analysis found three sterols—cholesterol, cholestanol, and epi-cholestanol—but no cholestanone was detected in their bodies (Appendix A). However, the concentrations of these detected sterols varied. There were significantly higher concentrations of epi-cholestanol (*t*_14_ = 8.86, *p* < 0.0001; Figure 6a) and cholesterol (*t*_14_ = 5.77, *p* < 0.0001; Figure 6c) but lower concentration of cholestanol (*t*_14_ = 7.23, *p* < 0.0001; Figure 6b) in *M. separata* than in *S*. *frugiperda*.

## 4. Discussion

Insects are sterol auxotrophs and entirely depend on an exogenous supply [26,31]. They tend to maintain constant cholesterol levels and adjust unsuitable sterols below a certain level, and the failure of this regulation can lead to death [13,36]. Therefore, disruption of the processes of sterol acquisition by changing plant sterol composition is a promising method for controlling many phytophagous pest species [29]. Genetically manipulated plants containing atypical phytosterols exhibit biological control potential [29,37,38]. In this study, for the first time, we explored the sterol requirements of two closely related lepidopteran pests, *M*. *separata* and *S*. *frugiperda*, as well as their metabolic characteristics on atypical steroids.

Cholesterol requirement experiments showed that the concentration of 1 mg/g dietary cholesterol allowed the good survival of both insects, but *S*. *frugiperda* seemed to be more tolerant to lower cholesterol concentrations, with a larger number of surviving individuals and a higher growth rate than *M*. *separata*. Insects tend to maintain constant cholesterol levels. For example, *D. melanogaster* attempted to maintain membrane sterol levels by reducing growth when sterol availability was restricted, with the failure of this regulation leading to death [13]. We found that many deaths occurred in *M*. *separata* and *S*. *frugiperda* larvae due to severe internal cholesterol deficiency. Therefore, compared to *M*. *separata*, the higher tolerance of *S*. *frugiperda* to sterol-deficient diets might result from their lower demand for cholesterol.

As 1 mg/g dietary cholesterol can support the growth of both insects, a diet containing the same concentration (1 mg/g) of cholestanol or cholestanone was used to evaluate the insects’ performance on atypical sterols. The performances of both *M*. *separata* and *S*. *frugiperda* larvae on the cholestanol diet were similar to those on the cholesterol diet, which indicated that cholestanol could support their growth as well as cholesterol. *Drosophila melanogaster* and *Caenorhabditis elegans* have been reported to use various cholesterol-like molecules, such as some phytosterols, in their lipid bilayers [39,40]. The flexibility to use other sterols as the “sparing” sterol for structural purposes to replace cholesterol was also demonstrated in some other lepidopteran insects when cholesterol (or sterols that can be converted into cholesterol) was deficient in food [41]. The “sparing” mechanism of using other sterols to form the membrane structure and employing spare cholesterol in hormone synthesis is believed to be highly beneficial [13,42].

Similar to the negative effect of cholestanone, a putative deleterious steroid, on many insects, cholestanone also showed remarkable lethality to *M*. *separata* [43,44]. In contrast, nearly 80% of the *S*. *frugiperda* individuals on cholestanone pupated. Moreover, *S. frugiperda* exhibited a better performance on cholestanone than *M*. *separata* in terms of food consumption and weight gain. Cholestanone may adversely affect the insect molting process by acting as a hormone mimic because this ketone has a similar structure to 3-dehydroecdysone (a precursor of ecdysone) [45]. To reduce the negative effects of atypical sterols, insects can convert harmful sterols into low-toxicity sterols/steroids, or selectively excrete them by ATP-binding cassette (ABC) transporters [46,47]. In this study, we checked the body sterol profiles of the two experimental insects. We found that there were substantial amounts of cholesterol, epi-cholestanol, and cholestanol in both species, but no cholestanone was detected. As we did not feed any epi-cholestanol or cholestanol, these two metabolites must be the metabolites of cholestanone. The different stereo-structures of these two isomers, epi-cholestanol and cholestanol, may be closely related to the enzymes involved in the conversion of cholestanone [48]. Notably, two other lepidopteran insects, *H*. *virescens* and *Manduca sexta*, can also convert cholestanone into the two isomers, and an extremely low amount of cholestanone was detected in them [31]. Therefore, lepidopteran insects seem to acquire a high efficiency in eliminating cholestanone by catalyzation and/or excretion. Moreover, *S*. *frugiperda* possessed significantly more cholestanol but less epi-cholestanol. A similar metabolic difference was also reported for *H. virescens* and *M. sexta*. In this study, we were able to correlate insect performance with cholestanone metabolism. The survival and pupation rates of both *M*. *separata* and *S*. *frugiperda* larvae on the cholestanol diet were similar to those on the cholesterol diet, which indicated that cholestanol could be used (or spared) by these two species. In contrast, epi-cholestanol cannot be used as a substitute for cholesterol in the cellular membrane because it lacks an equatorial 3-OH group, a stereo-structure that seems to be essential in the cellular membrane [25,47,49]. When we supplied *M*. *separata* with cholesterol in its diet, its performance recovered (Appendix A). Thus, the higher performance of *S*. *frugiperda* (in comparison to *M*. *separata*) on cholestanone is likely related to its preference in converting cholestanone into cholestanol, and its lower demand for cholesterol.

In this study, we unexpectedly found that *S*. *frugiperda* had an extremely high tolerance to cholestanone, and we further explored the difference in sterol metabolism and demand between *S. frugiperda* and *M*. *separata*. The results reflect the divergence in the sterol nutrition biology between these two closely related species, which could help us to better evaluate the efficiency of using phytosterol-manipulated plants to control insect pests.

## 5. Conclusions

Sterol usage in *S. frugiperda* and *M. separata*, two important global pests, were investigated for the first time in this study. Our results demonstrated that *S*. *frugiperda* and *M. separata* performed well on cholestanol but exhibited discrepancies toward a deleterious atypical sterol, cholestanone. The fall armyworm, *S. frugiperda*, exhibited an unexpectedly high tolerance to cholestanone, which caused extensive death in other lepidopteran pests, including in *M. separata*, as demonstrated in this study. We explored the reasons for this by comparing *S. frugiperda* with *M*. *separata* from a sterol metabolism and demand perspective. The results showed that *S*. *frugiperda* was superior to *M*. *separata* in terms of its efficiency when converting ketone into the available stanol, and its lower demand for cholesterol, which helped the pest to survive better when cholesterol was deficient. These results highlighted the metabolic characteristics and discrepancies of these two closely related species and provided basic information in relation to the potential for pest control using phytosterol-manipulated plants.

## Figures and Tables

**Figure 1 insects-13-00876-f001:**
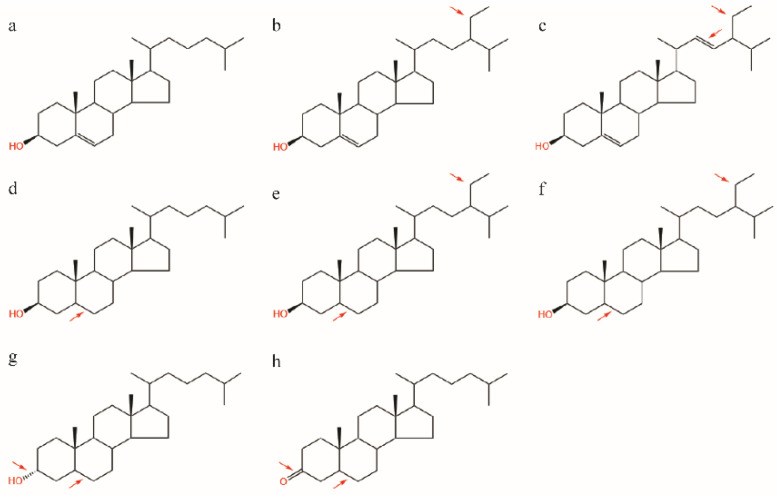
Chemical structure of sterols/steroids in this study. Cholesterol (**a**) is the dominant sterol in most animals; sitosterol (**b**) and stigmasterol (**c**) are the common phytosterols; cholestanol (5α-3β-ol) (**d**), sitostanol (**e**), stigmastanol (**f**), and epi-cholestanol (5α-3α-ol) (**g**) are the saturate stanols; cholestanone (**h**) has a ketone group in C3 instead of a hydroxyl and has no Δ5 double bond in the sterol nucleus compared with cholesterol. Red arrows indicate the differences between cholesterol and other sterols/steroids.

**Figure 2 insects-13-00876-f002:**
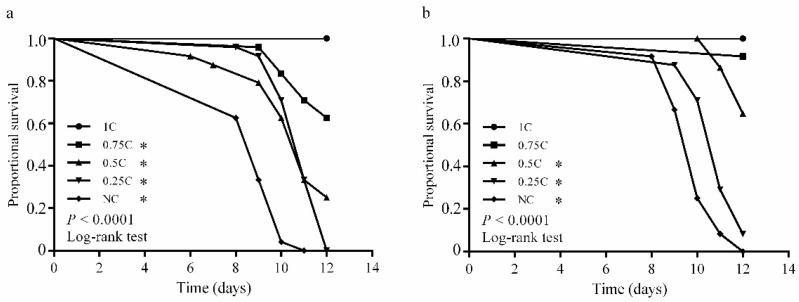
Larval survival of *Mythimna separata* (**a**) and *Spodoptera frugiperda* (**b**) on five cholesterol concentrations by the 12th day. * represents a significant difference in comparison to 1C at *p* < 0.05.

**Figure 3 insects-13-00876-f003:**
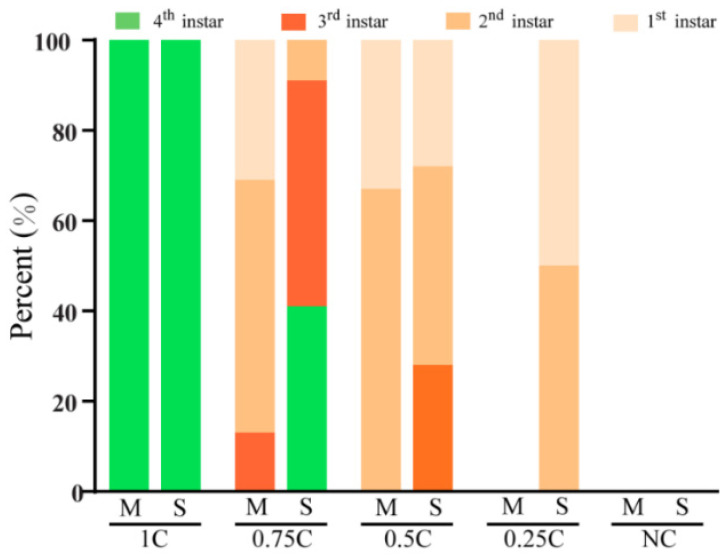
Proportions of larvae alive in each instar. Instar proportions of *Mythimna separata* (M) and *Spodoptera frugiperda* (S) larvae on diets containing different cholesterol concentrations on the 12th day.

**Figure 4 insects-13-00876-f004:**
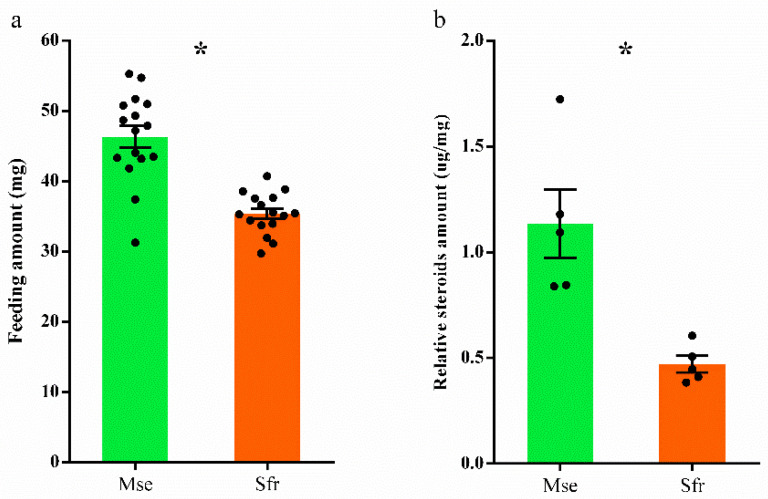
Performance of *Mythimna separata* (Mse) and *Spodoptera frugiperda* (Sfr) larvae on the diet containing 1 mg/g cholesterol. The feeding amount (**a**) was recorded during two stadia (the 4th and 5th instars). Upon reaching the 6th instar, the insects were sacrificed, and the body relative steroids amount (**b**) was measured by GC/MS technology. Cholesterol was the only sterol/steroid found in these samples. Data are calculated from five biological replicates and shown as means ± standard errors. * represents a significant difference at *p* < 0.05.

**Figure 5 insects-13-00876-f005:**
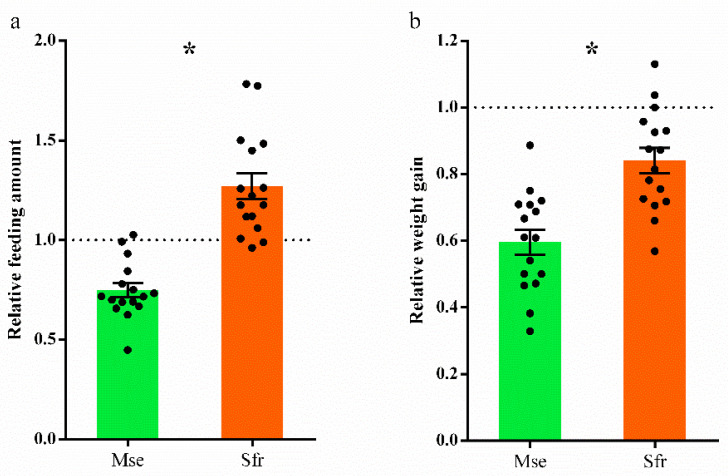
Larval performance of *Mythimna separata* and *Spodoptera frugiperda* on cholestanone. (**a**) Relative food intake and (**b**) relative weight gain during the 4th and 5th instars were presented. The dotted line represents where the insects consumed the same amount of food (**a**) or had the same level of weight gain (**b**) between the cholesterol diet and the cholestanone diet. Data are calculated from 16 biological replicates and shown as means ± standard errors. * represents a significant difference at *p* < 0. 05.

**Figure 6 insects-13-00876-f006:**
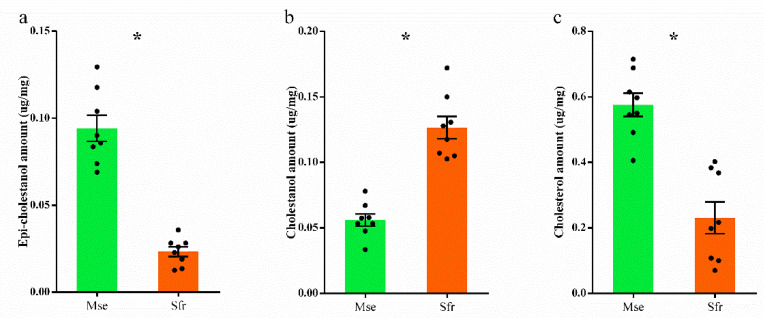
Body steroid profiles in *Mythimna separata* and *Spodoptera frugiperda* on cholestanone. Three sterols, (**a**) epi-cholestanol, (**b**) cholestanol, and (**c**) cholesterol were detected. Data are calculated from five biological replicates and shown as means ± standard errors. * indicates significant differences between *M*. *separata* and *S*. *frugiperda* at *p* < 0.05.

**Table 1 insects-13-00876-t001:** Performance of the two caterpillars reared on a diet containing cholesterol, cholestanol, or cholestanone at a concentration of 1 mg/g (dry mass). * in each row indicates significant differences compared to the cholesterol treatment at *p* < 0.05.

	*Mythimna separata*				*Spodoptera frugiperda*		
Dietary sterols	Cholesterol	Cholestanol	Cholestanone		Cholesterol	Cholestanol	Cholestanone
Development time (day)	23.21 ± 1.5	28 ± 1.32 *	—		21.17 ± 1.7	30.71 ± 1.65 *	31.67 ± 2.03 *
Pupation rate (%)	100	100	0 *		100.0	100.0	79.17 ± 13.82 *
Pupal mass (mg)	432.67 ± 17.63	395.88 ± 30.57 *	—		228.38 ± 11.43	202.83 ± 21.54 *	209.33 ± 18.45 *

## Data Availability

Not applicable.

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
