# Peer review of "Discrepancy in Sterol Usage between Two Polyphagous Caterpillars, Mythimna separata and Spodoptera frugiperda"

_insects, 2022, doi:10.3390/insects13100876_

Round 1

Reviewer 1 Report

Judging from the conclusions proposed in the article, the experiments involved are basically complete to support the conclusion, but there are still some minor flaws, so I put forward the following suggestions:

1. The Growth requirement of cholesterol experiment can be placed at the beginning to explain the choice of 1mg/g as the experimental treatment concentration or the reason for choosing 1mg/g as the experimental treatment concentration.

2. The description of the experimental method is as detailed as possible, explaining the reasons for choosing this material and method.

3. When describing the experimental results obtained, the words of summary can be added appropriately.

Minor sugesstion:

1, on line 127, "prism" should be "Prism".

2.  on line 183 and 184, cholestanol and cholestanone should be exchanged.

3.  Mythimna separata (Mse) and Spodoptera frugiperda (Sfr) on line 200 should be shown on line 186.

Author Response

Response to Reviewer 1 Comments

Point 1: The Growth requirement of cholesterol experiment can be placed at the beginning to explain the choice of 1mg/g as the experimental treatment concentration or the reason for choosing 1mg/g as the experimental treatment concentration.

Response 1: Thanks for pointing it out. We moved this experiment forward. We also modified the description in the revised manuscript. Please see Line 94-98.

Point 2: The description of the experimental method is as detailed as possible, explaining the reasons for choosing this material and method.

Response 2: We rephrased this section, and added more information.

Point 3: When describing the experimental results obtained, the words of summary can be added appropriately

Response 3: As suggested, we added the words of summary. Please refer to Line 157, 177.

Minor suggestion:

Point 1: on line 127, "prism" should be "Prism".

Response 1: Corrected. Please see Line 132.

Point 2: on line 183 and 184, cholestanol and cholestanone should be exchanged.

Response 2: We changed the position of cholestanol and cholestanone in Table 1, and now the sequecne of cholestanol and cholestanone was consistent. Please refer to Table 1.

Point 3: Mythimna separata (Mse) and Spodoptera frugiperda (Sfr) on line 200 should be shown on line 186.

Response 1: Corrected, thanks. Please refer to Line 198 in the revised manuscript.

Reviewer 2 Report

This paper investigated the tolerance of two different lepidopteran insects against atypical sterols. The data demonstrated the impact of atypical diet feeding for development and sterol composition in the larvae. Data is interesting, but data handling is not imperfect, and discussion is weak. This paper includes the finding about cholesterol lacking in two target species and uptake of noxious sterols. These two things are quite different physiological matter. If author discuss these two things in the same paper, additional experiment, i.e. experiment using diet with both cholesterol and atypical sterols should be required. Discussion should be connected these two matters, but insufficient in the manuscript. At least, te-construction of discussion will be need for accept.

Major revision

1. Most of discussion part is explanation about previous literatures. Such information should be included in the introduction part.

2. Fig. 2 showed that the feeding effect of 1K diet for two target species. Fig. 2a showed that M. separata feed less amount when they fed diet including cholestanone, whole Fig. 2b showed that they got heavier with the diet than feeding diet including cholesterol. These data look contradictory, but the explanation about the phenomenon was not included in the discussion part. Author need explanation about this point.

3. Author mentioned that cholestanone was not detected from larvae feeding with 1K diet. Does any data show that cholestanone could be incorporated in the midgut, or other tissues after transportation by lipoprotein in both M. separata and S. frugiperda? Is their possibility that failure of cholestanone detection come from immediate elimination instead of metabolism? Failure of cholestanone detection looks strange because they could feed a reasonable amount of diet including cholestanone. Author need explanation about this point.

4. Fig. 4 showed the larval survival of two species feeding with diet including insufficient cholesterol. How many larvae were used in the experiment? If the numbers of specimen were small, usage of % were not suitable. Instead, ratio (0.0 to 1.0) is better. Discussion about the cause of larval death is required. How about the case using diet including different concentrations of cholestanone? If authors had such information, it should be included in the manuscript. The information will help to consider the impact of cholesterol lacking (Fig. 4 ad 5), and uptake of noxious sterols to two species.

Author Response

Response to Reviewer 2 Comments

Point 1: Most of discussion part is explanation about previous literatures. Such information should be included in the introduction part.

Response 1: We moved some information into the introduction and revised the discussion accordingly. Please see Line 60-65. Thanks.

Point 2: Fig. 2 showed that the feeding effect of 1K diet for two target species. Fig. 2a showed that M. separata feed less amount when they fed diet including cholestanone, whole Fig. 2b showed that they got heavier with the diet than feeding diet including cholesterol. These data look contradictory, but the explanation about the phenomenon was not included in the discussion part. Author need explanation about this point.

Response 2: We redrew this figure (Figure 2), and it was ordered as Figure 6 in the revised manuscript. As these two species have different body size, we calculated the relative food intake and relative weight gain. Relative food intake was represented by the ratio of the food intake of the insect on cholestanone to the food intake on cholesterol. Relative weight gain was represented by the ratio of the weight gain of the insect on cholestanone to that on cholesterol. M. separata fed relatively less cholestanone diet and got relatively less weight, respectively. Please refer to Line 177-180.

Point 3: Author mentioned that cholestanone was not detected from larvae feeding with 1K diet. Does any data show that cholestanone could be incorporated in the midgut, or other tissues after transportation by lipoprotein in both M. separata and S. frugiperda? Is their possibility that failure of cholestanone detection come from immediate elimination instead of metabolism? Failure of cholestanone detection looks strange because they could feed a reasonable amount of diet including cholestanone. Author need explanation about this point.

Response 3: We have no evidence of the immediate elimination of cholestanone yet, but it is possbile. Moreover, it is very clear that cholestanone can be converted into cholestanol and epi-cholestanol. Failure of the detection is not strange as extremely low amount of cholestanone was detected in other two species feeding on a large amount of cholestanone in a previous study. We rephrased this part in the discussion. Please refer to Line 256-258, 260-262, and 263-267.

Point 4: Fig. 4 showed the larval survival of two species feeding with diet including insufficient cholesterol. How many larvae were used in the experiment? If the numbers of specimen were small, usage of % were not suitable. Instead, ratio (0.0 to 1.0) is better. Discussion about the cause of larval death is required. How about the case using diet including different concentrations of cholestanone? If authors had such information, it should be included in the manuscript. The information will help to consider the impact of cholesterol lacking (Fig. 4 ad 5), and uptake of noxious sterols to two species.

Response 4: We revised the survival in the Figure 2. Besides, we added other data (Figure S4), and the results showed that if cholesterol (1 mg/g) was suppled with cholestanone, M. separata grew well. Therefore, the higher performance of S. frugiperda (in comparison to M. separata) on cholestanone is likely related to its preference in converting cholestanone into cholestanol, and the lower demand for cholesterol. Please refer to Line 275-278.

Round 2

Reviewer 2 Report

The manuscript has been much improved. However, authors need one more revision.

Minor revision

1.     Fig. 4: The meaning of “relative feeding amount” is still unclear. What does the ratio “1.0” mean? The explanation how the data was standardized should be included in the figure legend.

2.     The numbers of references must be corrected. Some references were lacking in the reference section. References after No. 45 were lacking in the reference section but included in the manuscript.  

3.     The manuscript would be improved by a English revision before publication. i.e. ”M. separata larvae were found consumed much more food… (Line197)”.

4.     As similar issue of above, the manuscript would be improved with careful check. It is a minute thing; species name is often written in as full name. These things make an impression that the manuscript is carelessly.

Author Response

Response to Reviewer 2 Comments

Point 1: Fig. 4: The meaning of “relative feeding amount” is still unclear. What does the ratio “1.0” mean? The explanation how the data was standardized should be included in the figure legend.

Response 1: We added more details in the method. Please see Line 121-132, 216-217. Because adding these words would make the legend very lengthy, we prefer to keep them in the method section. Thanks.

Point 2: The numbers of references must be corrected. Some references were lacking in the reference section. References after No. 45 were lacking in the reference section but included in the manuscript. 

Response 2: We carefully checked the version we submit, and there are 4 references after No. 45. We guess you may somehow miss these four references. Please see Line 451-458. Anyway, we have double checked these references. Thanks for your careful check.

Point 3: The manuscript would be improved by a English revision before publication. i.e. ”M. separata larvae were found consumed much more food… (Line197)”.

Response 3: We have asked a senior entomologist to help us edit the whole manuscript, and the changes were highlighted. Hopefully, it can meet the standard now. Thanks.

Point 4: As similar issue of above, the manuscript would be improved with careful check. It is a minute thing; species name is often written in as full name. These things make an impression that the manuscript is carelessly.

Response 4: It it the general rule to avoid starting a sentence with an abbreviation in formal and academic writing. We think it should be also applicable to the scientific names. To avoid this confusion, we have restructured these sentence and no scientific name appear in the first place of a sentence. Please see Line 69, 70, 82, 162, 238, 296. Thanks for you advice.
